# Nitric Oxide: The Missing Factor in COVID-19 Severity?[note 1]

**DOI:** 10.3390/medsci10010003

**Published:** 2021-12-23

**Authors:** Alexandros Nikolaidis, Ron Kramer, Sergej Ostojic

**Affiliations:** 1NKG Pharmaceuticals, Signal Hill, CA 90755, USA; 2ThermoLife International, Phoenix, AZ 85048, USA; ron@thermolife.com; 3Applied Bioenergetics Lab, Faculty of Sport and PE, University of Novi Sad, 21102 Novi Sad, Serbia; sergej.ostojic@chess.edu.rs

**Keywords:** COVID-19, SARS-CoV-2, nitric oxide, NO, comorbidities, etiology

## Abstract

Coronavirus disease 2019 (COVID-19) is a contagious respiratory and vascular disease that continues to spread among people around the world, mutating into new strains with increased transmission rates, such as the delta variant. The scientific community is struggling to discover the link between negative COVID-19 outcomes in patients with preexisting conditions, as well as identify the cause of the negative clinical patient outcomes (patients who need medical attention, including hospitalization) in what seems like a widespread range of COVID-19 symptoms that manifest atypically to any preexisting respiratory tract infectious diseases known so far. Having successfully developed a nutritional formulation intervention based on nitrate, a nitric oxide precursor, the authors hypothesis is that both the comorbidities associated with negative clinical patient outcomes and symptoms associated with COVID-19 sickness are linked to the depletion of a simple molecule: nitric oxide.

## 1. Prologue

Coronavirus disease 2019 (COVID-19) is a contagious respiratory and vascular disease caused by infection with the coronavirus SARS-CoV-2 [1]. While many patients have mild symptoms, a fraction of them develop acute respiratory distress syndrome (ARDS) possibly precipitated by a cytokine storm, multi-organ failure, septic shock, and blood clots [2,3]. Increased severity of diseases is commonly associated with a persistent drop in blood oxygen saturation levels (SpO_2_). Longer-term organ damage (in particular, the lungs and heart [4]) has been observed, and there is a growing concern about a significant number of patients who have recovered from the acute phase of the disease but continue to experience a range of effects, referred to as post-COVID-19 conditions or long-COVID [5].

A characteristic of COVID-19 infection severity is the almost linear association with age, with most of the severe and fatal cases occurring in the elderly, while children, especially those under the age of 11, remain relatively unscathed by the pandemic [6] (Figure 1). 

Many comorbidities have been associated with negative and severe COVID-19 clinical outcomes. Among them, the most prevalent are hypertension, metabolic disease (mainly diabetes), heart disease, obesity (which is a major risk factor in the development of metabolic disease, hypertension, cardiovascular disease [7,8,9]) and asthma (Figure 2).

To date, the link between these risk factors and COVID-19 severity remains unknown, although many theories have emerged including the lower amount of ACE receptors in children, which increases with aging, being one of the first suggested and explored [10]. However, the theory did not hold ground and therapeutic approaches based on ACE inhibitors have failed to produce consistent clinical results [11]. Fueled by encouraging clinical data [12], the authors propose the theory that a key molecule is tied to the severity of COVID-19: nitric oxide.

## 2. About Nitric Oxide

Nitric oxide (NO) is a free radical gas that transmits signals in the organism. Pharmacology professor Lou Ignarro first discovered that this small molecule, thought to be an atmospheric pollutant, can have effects, such as vasodilation on biological organisms [13]. NO is produced by three isoforms of the NO synthase (NOS; EC 1.14.13.39), neuronal NOS (nNOS, NOS I), inducible NOS (NOS II) and endothelial NOS (eNOS, NOS III) (Figure 3). Of particular interest for this article, the third NOS isoform, endothelial NOS (eNOS, NOS III), is mostly expressed in endothelial cells. NO from eNOS keeps blood vessels dilated, controls blood pressure, and has numerous other vasoprotective and anti-atherosclerotic effects. Many cardiovascular risk factors lead to oxidative stress, eNOS uncoupling, and endothelial dysfunction in the vasculature [14]. NO has also been implicated in the development of diseases, including gastrointestinal cancer due to the formation of nitrosamines after reaction with amine groups (such as those in free amino acids) [15]. Another major stepstone in NO research has been the work of Jon Lundberg and Eddie Weitzberg, who discovered that nitrate (NO_3_^−^) and nitrite (NO_2_^−^), at the time thought to be inert byproducts of NO metabolism and contaminants in food, can function as a secondary source of NO in the body, especially in cases of hypoxia (Figure 4) [16]. Interestingly, nitrate and nitrite can be found in some types of vegetables—such as conventionally grown beetroot and leafy vegetables—and when ingested they can be reduced by either bacterial activity on the gastrointestinal tract or enzymes in the blood and tissues to generate NO with robust beneficial effects, such as the lowering of blood pressure and improvements in mitochondrial efficiency, which result in reduced oxygen consumption that can in turn increase athletic performance [17,18].

Since these discoveries by Drs Lundberg and Weitzberg, a spur of interest in researching the nitrate–nitrite–NO pathway has commenced, with hundreds of clinical trials studying its effects in the body and the potential to treat and prevent disease.

### 2.1. Nitric Oxide and COVID-19

The correlation between NO levels and respiratory viral disease severity was first noted by Pedja et al., which reported that levels of NO oxidation products in serum were higher in patients that survived acute respiratory distress syndrome caused by H1N1 [19], in comparison with non survivors. H1N1 infection is also known to increase exhaled NO levels. [20]. Later on, following the emergence of SARS-CoV’s predecessor, SARS-CoV-1, Akerstorm et al. found in in vitro experiments that incubating cells with the direct NO donor S-nitroso-N-acetylpenicillamine (SNAP) could inhibit the replication of the virus [21]. Successful containment of the SARS-CoV-1, that resulted in a much more lethal disease than COVID-19, led to the abandonment of further research of this therapeutic pathway for the treatment of SARS. In further in vitro tests in 2020, Akaberi et al. confirmed that NO procured from SNAP can delay or completely prevent the development of SARS-CoV-2 viral cytopathic effect in treated cells and also that the observed protective effect correlated with the level of inhibition of viral replication [22]. Midway through the COVID-19 pandemic, inhaled NO gas emerged as a possible therapeutic process that could increase patient oxygenation and potentially lead to better clinical outcomes [23]. Possible therapeutic mechanisms include pulmonary vasodilation, antimicrobial and antiviral effects, reduction of pulmonary hypertension, increase in ventilation and bronchodilation, increase of blood flow in ventilated lung units, and anti-inflammatory and antithrombotic properties.

After inhalation, NO diffuses rapidly across the alveolar–capillary membrane into the subjacent smooth muscle of pulmonary vessels to activate soluble guanylyl cyclase. This enzyme mediates many of the biological effects of NO and is responsible for the conversion of GTP to cGMP [24] (Figure 5).

This therapeutic approach (inhaled NO) suffers from quite a few drawbacks, however, the main one being its availability and high cost, which is over $100 per hour for hospital use. With the average hospitalized patient requiring 30–40 h of inhaled NO therapy, this costly approach is reserved only for severe cases. There have also been emerging problems due to the potential toxicity of NO that can, among others, cause methemoglobinemia, which lowers the blood’s ability to carry oxygen [25]. Another majorly concerning issue with current NO gas therapies is that in the presence of oxygen, NO can convert to nitrogen dioxide (NO2), a toxic gas with no therapeutic value [26]. Prolonged contact of NO2 with the lung epithelial lining fluid can lead to edema, bronchoconstriction, and reduced forced expiratory volume [27]. This requires constant monitoring of the patients administered inhaled NO gas, which preferably must be performed by a practitioner experienced in NO therapy and trained medical staff. For these reasons, as well as the fact that NO therapy is reserved for COVID-19 patients at an advanced stage of COVID-19-induced lung damage, despite a relatively consistent initial beneficial effect of inhaled NO gas in improving oxygen saturation levels, it has come under question whether current methods of administering NO gas therapy can actually improve mortality rates in hospitalized COVID-19 patients [28].

### 2.2. Nitric Oxide and Age

It is well established through human clinical trials that NO synthase expression, especially eNOS, declines with age, which leads to a drop into serum NO metabolite levels [29], and that the consequent deficiency in NO levels can negatively impact the progression of geriatric diseases [30]. McCarti confirmed that NOx decreases with age among both male and female subjects. There was no significant difference in serum NOx levels between male and female subjects in all the age groups. It was observed that from 60 (group V) onwards, the decrease of NOx concentration between male and female subjects remained near similar.

As endothelial synthase activity declines with age, it contributes to the deterioration of the cardiovascular system and all organs dependent on it [31].

Aging is associated with a major increase in mortality rates as well as severe clinical outcomes in a multitude of diseases, including viral infections such as COVID-19, with children generally experiencing much milder infection symptoms than adults [32]. One could easily see a correlation of declining NO levels with disease severity and mortality rates. However, while it is well known that correlation does not necessarily equal causation, and that just this one link could be overlooked as a mere coincidence, it is the correlation with all the other major COVID-19 comorbidities and low levels of nitric oxide that may establish a more than coincidental link.

### 2.3. Comorbidities of COVID-19 and Nitric Oxide

#### 2.3.1. Nitric Oxide and Hypertension/Cardiovascular Disease

Hypertension is the number one comorbidity associated with COVID-19 hospitalizations and severe clinical outcomes such as death. Due to the extremely short physiological half-life of this gaseous free radical, alternative strategies for the detection of reaction products of NO biochemistry have been developed. The quantification of NO metabolites in biological samples can provide valuable information with regards to in vivo NO production, bioavailability, and metabolism [33]. One such common method is the quantification of NO’s primary metabolites total levels in the blood, nitrate and nitrite, the sum of them being commonly referred to in the literature as NOx. Nitrite is a more sensitive and indicative marker of eNOS compared to other nitric oxide synthases, as eNOS stimulation leads to rapid increases in plasma nitrite levels while nitrate levels remain relatively unchanged [34]. The association between lower NOx levels and hypertension prevalence is well established [35]. In a large study involving 2968 subjects, serum NOx values were negatively correlated with systolic blood pressure ≥160 mmHg in men (r = −0.523, *p* = 0.002). Serum NOx was higher in men with stage 1 hypertension, while those with stage 2 hypertension had significantly lower NOx levels. In men, medication increased serum NOx concentration in both stages of hypertension, but in women, a significant increase was observed only in stage 1 hypertension. Dietary inorganic nitrate has been utilized with success to lower blood pressure and maintain healthy blood pressure levels with a growing amount of evidence to support it [36]. Furthermore, it has been well established that hypoxia also raises blood pressure, as the body struggles to pump more blood to provide the necessary oxygen to the tissues [37]. Thus, it is entirely plausible that when COVID-19 infection advances to ARDS and causes hypoxia as a consequence, it also raises blood pressure. This can lead to a vicious cycle as hypertension induced by NO deficiency and hypoxia thus increase COVID-19 severity, which in turn causes a greater rise in blood pressure. It is also noted that the stress induced in COVID-19 patients with diagnosed hypoxia and breathing difficulties can also raise blood pressure [38].

Due to the close relationship of NO with hypertension and endothelial health, it is not surprising that NO levels are closely tied to cardiovascular disease prominence. NO plays an important role in the protection against the onset and progression of cardiovascular disease [39]. 

For example, NO deficiency can promote endothelial dysfunction, atherosclerosis, hypertension, decrease in heart contractility, platelet adhesion and aggregation, and inflammation. On the other hand, sufficient NO availability is vital for angiogenesis, which can alleviate ischemia in certain tissues. 

Among the various risk factors for CVD, high blood pressure (BP) is the one with the strongest evidence for causation, and it has a high prevalence of incidence in CVD patients [40]. COVID-19 infection has been found to both accelerate endothelial dysfunction and result in NO deficiency [41].

#### 2.3.2. Nitric Oxide and Diabetes

The second most common comorbidity associated with severe COVID-19 outcomes is diabetes, both type I and type II [42]. Diabetes, especially type II, is closely associated with obesity, another COVID-19 risk factor [43]. NO synthesis is known to be impaired in diabetes type II patients with renal disease [44]. Dysfunction of eNOS is closely associated with the development of diabetic nephropathy [45]. Chronic increase in blood glucose levels is known to inhibit eNOS activity and procure a reduction of NO levels [46]. NO bioavailability is decreased in animal models with diet-induced obesity and in obese and insulin resistant patients, and increasing NO output has remarkable effects on obesity and insulin resistance [47]. Inorganic nitrate and nitrite have been proposed as emerging therapeutic approaches to diabetes in a limited number of studies [48]. Surprisingly enough, nitrate supplementation in diabetic patients resulted in significantly lower 3-nitrotyrosine levels, possibly due to an antioxidant effect of inorganic nitrate [49].

#### 2.3.3. Nitric Oxide and Metabolic Disease/Obesity

Metabolic disease is a group of diseases and disorders that disrupt normal metabolism, the process of converting food to energy on a cellular level. Metabolic syndrome is a condition cluster of these conditions that occur together, which can in turn increase the risk of heart disease, stroke, and type 2 diabetes. Obesity, while not considered a disease by itself, is a consequence of many metabolic diseases and a prominent risk factor for type II diabetes, heart disease, and stroke [50]. Nitrate supplementation affects obesity and metabolic disease by enhancing the expression of brown adipose tissue specific genes, contributing to AMPK activation and GLUT4 translocation, augmenting mitochondrial fission in PKA-dependent and NO-independent manner, and attenuating oxidative stress via reduced NADPH oxidase activity [51].

#### 2.3.4. Nitric Oxide and Pregnancy

Pregnancy has been suggested as a risk factor in COVID-19 severity. Pregnant women are more at risk of contracting COVID-19 due to their weakened immune system [52]. In a 100 people case study, NO levels were found to be lower in pregnant women [53]. Gestational hypertension and gestational diabetes are also a common occurrence in pregnant women, with a higher chance of co-occurring, and both can increase the severity of COVID-19 [54]. A recent study has found that red blood cells from women with preeclampsia cause endothelial dysfunction ex vivo due to a deficiency in NO that can be reversed by treatment of the cells with sodium nitrite solution [55].

#### 2.3.5. Nitric Oxide and Immune Suppression

NOS2/iNOS was originally described as an enzyme that is expressed in activated macrophages and that generates nitric oxide (NO) from the amino acid L-arginine, thereby contributing to the control of replication or killing of intracellular microbial pathogens. Since interferon (IFN)-gamma is the key cytokine for the induction of NOS2 in macrophages and the prototypic product of type 1 T-helper cells, a high-level expression of NOS2 has been regarded to be mostly restricted to the adaptive phase of the immune response [56]. NO is a key regulator of myeloid inflammatory cell apoptosis. It possesses both anti- and proapoptotic properties, depending on the concentration of NO and the source from which NO is derived [57]. Nitric oxide is also essential in macrophage activation in response to infection and to the body’s whole immune response [58],—macrophage cells release NO in response to inflammatory signals—and the NO increase in macrophages results in a multitude of changes in macrophage immunometabolism with multiple implications in COVID-19 patients [58]. For example, supplementation with the NO precursor arginine in COVID-19 patients has been demonstrated to suppress proinflammatory cytokine by mononuclear cells, which include macrophages [59]. In sepsis (a condition that shares many pathophysiological and clinical features with COVID-19 [60]) the iNOS nitric oxide synthesis is dysregulated with exaggerated production, leading to cardiovascular dysfunction, bioenergetic failure, and cellular toxicity whilst at the same time impaired microvascular function may be driven in part by the reduced nitric oxide synthesis by the endothelium [61]. During sepsis, iNOS is induced by various cell types, including immune cells, endothelial cells, as well as myocytes, in response to endogenously generated inflammatory mediators, such as cytokines, platelet products, superoxide anions, and NO itself [62]. NO generation from iNOS has been shown to suppress lymphocyte proliferation, and this might explain the exhaustion of lymphocytes in severe COVID-19 [63,64]. Ex vivo enhancement of septic patient’s platelet and endothelial cell eNOS activity reduces sepsis-related neutrophil–endothelial cell interactions and may play a role in maintaining microvascular patency during septic shock [65]. Thus, eNOS malfunction would increase the severity of iNOS-induced deleterious effects, while its restoration could exhibit a protective effect.

#### 2.3.6. Nitric Oxide and COPD

Chronic obstructive pulmonary disease (COPD) is a risk factor for severe COVID-19 that leads to hospitalization and ICU admission [66]. Dysregulation of the endothelial nitric oxide pathway is associated with airway inflammation in COPD [67], while defective eNOS polymorphisms have been associated with increased oxidative stress [68]. Smoking and COPD are also common factors contributing to the malfunction of endothelial function, which can be measured as an increase in plasma 3-Nitrotyrosine levels (an NO catabolite) that is a marker of oxidative stress and inflammation [69,70].

#### 2.3.7. Nitric Oxide and Asthma

Asthma is an interesting case as it is well documented that asthmatics have increased iNOS expression and exhaled NO levels [71]. In fact, exhaled NO is a common diagnostic method for asthma [72]. However, animal studies have indicated that enhancing plasma NO production/availability via eNOS upregulation can reduce asthma severity [73]. It has also been found that asthmatic patients have a higher prevalence of dysfunctional eNOS [74]. Interestingly, children with asthma exhibit a higher level of plasma NOx compared to healthy children, regardless of whether their asthma is well controlled or not [75].

#### 2.3.8. Nitric Oxide and Smoking

Smoking has been found to decrease platelet NO synthesis in smokers [76]. Low platelet NO synthesis in smokers may result in the augmentation of platelet aggregation and thrombus formation, developing into acute coronary syndromes. Smoking can also decrease eNOS NO synthesis, which contributes to its deleterious effects in cardiovascular health, lung health, and lung infection severity [77].

## 3. Severe COVID-19 Clinical Outcomes

### 3.1. Is COVID-19 Simply a Case of Systematic NO Depletion?

NO exhibits antiviral activity on a variety of viral infections and one of the main mechanisms is the S-nitrosylation of viral proteins [78]. NO depletion has been suggested as a mechanism by which these viruses cause endothelial dysfunction, such as the HIV virus [79], which on animal models happens without a marked decrease in eNOS expression [80], suggesting a scavenging effect.

As mentioned above, Akerstorm et al. identified in 2005 NO as a potential inhibitor of the coronavirus SARS-CoV-1 replication. In later work [81], they suggested that the inhibition possibly was a result of inhibition by NO of two viral replication synthesis mechanisms. First, NO resulted in the reduction of the palmitoylation of nascently expressed spike (S) protein, which affects the fusion between the S protein and its cognate receptor, angiotensin converting enzyme 2 (ACE2). Secondly, NO treatment of the virus resulted in a reduction in viral RNA production in the early steps of viral replication, and this could possibly be due to an effect on the two cysteine proteases encoded in Orf1a of SARS-CoV-1 (Figure 6).

Using a similar methodology, with SNAP as the in vitro NO donor, Akaberi et al., noted in their SARS-CoV-2 in vitro experiments that NO could inhibit SARS-CoV-2 3CL recombinant protease in vitro and that the observed reduction in the SARS-CoV-2 protease activity was consistent with the S-nitrosation of the enzyme active site’s cysteine molecule residue (CYS145). Other cysteine (Cys) residues of clinical interest exist in various other proteins of SARS-CoV-2, such as in the critical for cellular invasion spike protein. Hati et al. found in in vitro experiments that in oxidant stress conditions, proximate Cys residue pairs of both the virus’s and ACE2’s sulfhydryl(-SH) would convert to disulfide groups (-S-S-) and such a conversion would greatly enhance the binding affinity between the virus’s spike protein and the host cell’s ACE2 [82] (Figure 7). It is quite possible biochemically that with sufficient NO availability the -S-S- formation could be prevented altogether by the S-nitrosation of the relevant sites.

NO is a very reactive and short-lived species, with thiol groups (-SH), such as those found in cysteine, being a primary target for its cellular signaling [83,84,85] (Figure 8).

In the endothelium, under normal conditions, such thiol groups are regenerated by the body’s own antioxidant mechanisms, but the system has been known to be prone to depletion, such as in the case of tolerance build-up to NO-based vasodilators, including nitroglycerin and similar drugs (isosorbide dinitrate, sodium nitroprusside) due to thiol group depletion [85]. Supplementation with -SH-containing compounds such as N-acetylcysteine (NAC) have in the past been utilized successfully to ameliorate this issue [86], but it is of concern that such products are gradually being pulled off the market as they fail to comply with regulatory organizations like the FDA. The disulfide bridges on the virus’s surface could also be a molecular target for NO that would lead to further NO depletion-cleavage of the -S-S- bond by NO, and subsequent formation of nitrosothiols has been documented both in laboratory chemistry experiments [87] and in in vitro and in vivo models [88]. It is a common principle in chemistry that a molecule that has reacted to form a new compound is a molecule lost. Thus, NO molecules “lost” by reaction with SARS-CoV-2 thiols (or other possible viral molecular targets such as disulfides) are lost to the body forever. As the disease progresses, it is estimated that 1–100 billion virions exist in the body of infected persons with a calculated theoretical total mass of up to 0.1 mg [89]. The concentration of NO in various tissues is in the nanomolar region; for example, it has been calculated that blood contains a maximum of 0.36 ng/L of NO [90], which assuming a total blood volume of 5 L would equal 1.8 ng for an average person. With the number of virions present in the blood, each one with 24–40 randomly arranged spike proteins, a significant reduction in plasma NO concentration due to scavenging from SARS-Cov-2 thiols does not seem implausible. That would be deleterious to people already suffering from NO deficiency, further exacerbating their health condition.

Due to the evidence outlined here, the authors propose that it is entirely in the realm of possibility that as the virus multiplies, the various thiol groups act as scavengers of produced NO, restricting the availability of produced NO, which, especially in a NO-deficient subject, would soon throw the entire system off balance into to what we characterize as severe COVID-19 disease, wherein the severity of infection corresponds to a direct correlation of the depletion in NO levels.

Other data exist to support such a tantalizing hypothesis. COVID-19 has, of course, some of the usual pathogenesis of respiratory disease viruses: inflammation caused by the body’s immune response, cough, fever, etc. Of unknown causality however are some clinical manifestations unique and very common to negative COVID-19 clinical outcomes: how harsh the virus is to the cardiovascular system (CVS); the formation of blood clotting; ARDS, which can lead to low oxygen saturation that can sometimes not be reversed by oxygen therapy; and anosmia (loss of smell).

N(G)-Nitro-L-arginine methyl ester (L-NAME) is a molecule used to study NO synthesis inhibition, causing a depletion in the body’s NO levels both in animal and human models. It is also used as a primary method to induce hypertension in animal models. While studies using L-NAME in humans to cause severe NO depletion and measure its effects are lacking, a look at some in vitro and animal studies provides more evidence for a possible connection.

Rubini et al. [91] found that NO synthesis inhibition by intraperitoneal L-NAME administration increases airway resistance in rats, resulting in increased work of breathing (the amount of energy needed by the respiratory muscles to produce enough ventilation and respiration to meet the metabolic demands of the body).In vitro, L-NAME treatment of human whole blood resulted in increased clotting and fibrinogenesis, a result consistent with the inhibitory effects of NO on platelet function and of the platelet-aggregating properties of NOS inhibitors [92]. Angelis et al. found that prolonged administration of L-NAME to rats through drinking water causes impairment in cardiac output, as well as drops in venous oxygen saturation during exercise versus untreated animals [93]. Nasal NO levels have been shown to have a connection with olfactory senses, as an airborne messenger and as an anti-infectious agent in the nose and sinuses, and to contribute to the mucociliary clearance [94]. Thus, it can be safely said that both in vitro and in vivo rat experiments using L-NAME to cause systemic nitric oxide depletion support the notion that the pathogenesis of COVID-19 is tied to, at least some degree, systemic NO reserves depletion.

### 3.2. Nitric Oxide and COVID-19 Patient Outcomes: What Does the Evidence Suggest?

Since the conception of the authors’ hypothesis that COVID-19 severity and pathology is correlated to NO bioavailability and successful treatment and prevention options can be developed manipulating this pathway, clinical evidence supporting this hypothesis has emerged. By comparing the NOx levels of 66 COVID-19 patients with those of 33 controls, Dominic et al. found significantly lower levels of NOx in COVID-19 patients vs. control (418.84 ± 153.03 nM vs. 286.69 ± 140.39 nM, *p* < 0.0001), as well as significantly lower plasma nitrite (free nitrite (292.63 ± 141.67 nM vs. 179.945 ± 164.0 nM, *p* = 0.0017) and bound S-Nitrosothiol fractions (243.19 ± 91.60 nM vs. 152.89 ± 85.39 nM, *p* < 0.0001) [95]. Wang et al. measured the nitrate and nitrite levels of 109 recovered adult COVID-19 patients 4 months after recovery and compared them to those of 166 uninfected adults. Lower nitrite and nitrite/nitrate, and higher nitrate levels were found among the recovered patients, indicating according to the authors long-lasting eNOS damage after COVID-19 infection [96]. Fasted nitrate levels have been used as an indicator of eNOS function, but since the work of Lauer et al., multiple studies have found that circulating nitrite and nitrate/nitrite ratios are a more accurate indication of eNOS dysfunction; thus, the lower nitrite levels in both studies are a clear indicator of eNOS dysfunction [97]. Since the authors in Wang et al. did not measure NOx levels of the patients before infection, it is unclear if eNOS dysfunction preexisted in the patients, was caused by COVID-19, or both. Future research for the utilization of NOx as predictors of COVID-19 severity as well as new therapeutic approaches focused on NO are necessary.

## 4. Conclusions

What started as an observation of the correlation between the severity of COVID-19 symptoms associated with old age, and the age-related decline of NO levels, has now tied low NO levels to all major high-risk groups of COVID-19 infection. This overwhelming amount of correlation data and the sound biochemical reasoning behind low NO levels and the corresponding conditions that increase COVID-19 infection severity as well as the symptomatology exclusive to COVID-19, the successful utilization of NO gas as a therapeutic option, and the emergence of evidence tying NOx to COVID-19 severity, make further research on the matter warranted. Moreover, the authors would like to suggest to people belonging to high-risk COVID-19 groups to take some easy and safe nutritional steps to improve their nitric oxide status, such as adding in their nutrition or supplementation regimen, quality sources of inorganic nitrate (such as conventionally grown beets and leafy vegetables, as organically grown vegetables contain little to no nitrate), which acts as a precursor of NO [98], or other dietary supplements delivering inorganic nitrates. Generally, they should aim for a daily dose of at least 300 mg nitrate (optimally 400 mg) and also at least 1 mg of folic acid, preferably in the form of 5-MTHF (a form effective for people suffering with 5,10-Methylene-tetrahydrofolate reductase deficiency), which can revert dysfunction of eNOS [99]. On a separate published peer-reviewed publication, the authors have explored the usage of a therapeutic nitrate-based formula with promising and unexpected results in improving SpO2 levels and relieving symptoms of patients recovering from COVID-19 [12].

## Figures and Tables

**Figure 1 medsci-10-00003-f001:**
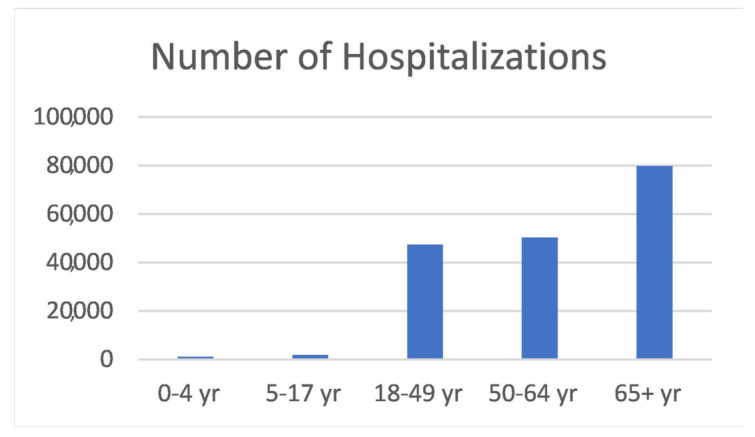
Number of COVID-19-associated hospitalizations for the period of 7 March 2020–22 May 2021 in the USA, categorized by age group. Source: US CDC https://gis.cdc.gov/grasp/covidnet/COVID19_5.html, accessed on 5 June 2021.

**Figure 2 medsci-10-00003-f002:**
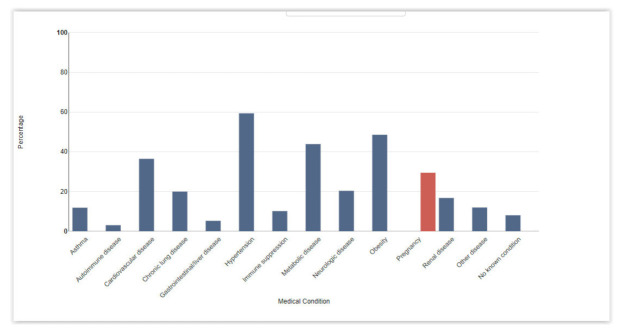
Underlying medical conditions and pregnancy by percentage in patients with COVID-19-related hospitalizations in the USA. Data are restricted to cases reported during 1 March 2020–31 March 2021, due to delays in reporting.

**Figure 3 medsci-10-00003-f003:**
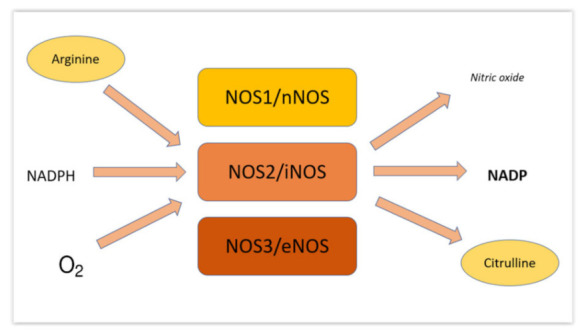
Biological pathways of nitric oxide generation in vivo in the body. Image created by authors.

**Figure 4 medsci-10-00003-f004:**
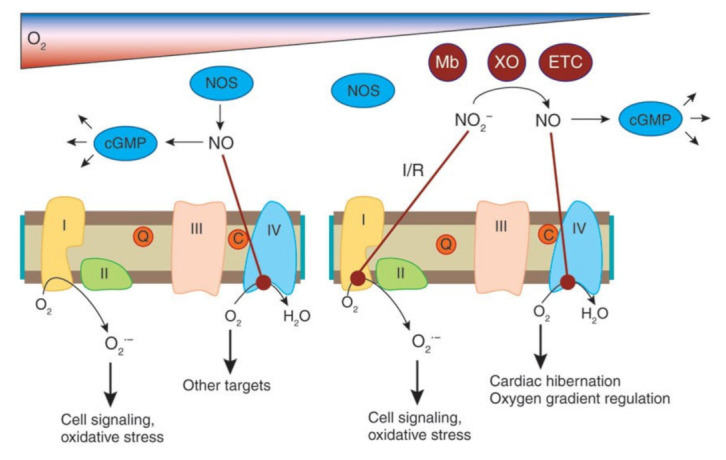
Molecular targets and biological effects of Nitric Oxide. Source: Lundberg et al., The nitrate-nitrite-nitric oxide pathway in physiology and therapeutics. Image reproduced with author’s and Nature publishing group’s permission.

**Figure 5 medsci-10-00003-f005:**
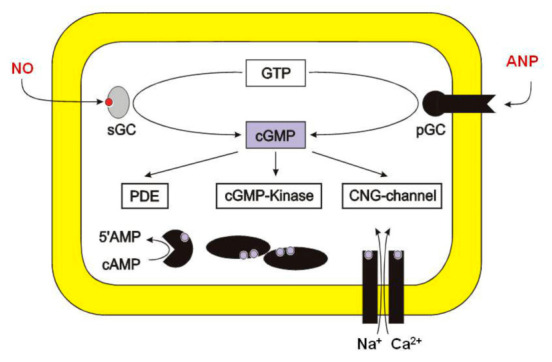
Activation of soluble guanylyl cyclase by nitric oxide. Source: https://commons.wikimedia.org/w/index.php?curid=15680898, accessed on 22 June 2021.

**Figure 6 medsci-10-00003-f006:**
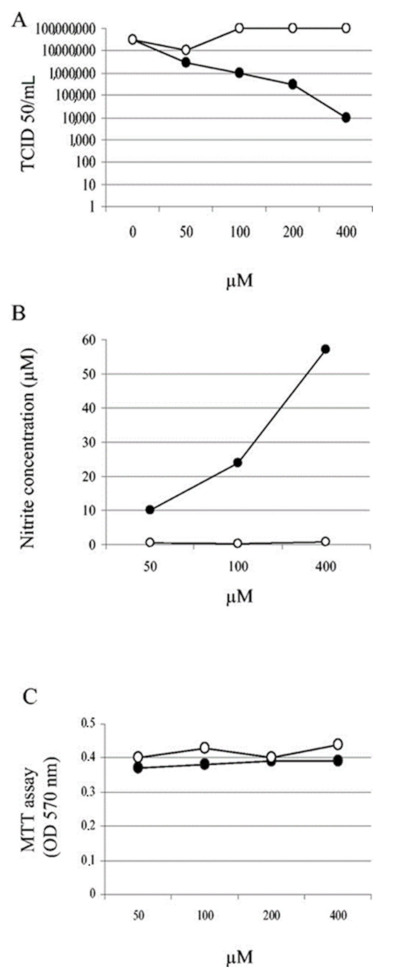
Cells were infected with SARS-CoV-1 at an MOI of 1.0. At 1 hpi, the cells were treated with different concentrations of SNAP (•) and NAP (○). (**A**) Supernatants were harvested at 24 hpi and titers were determined. (**B**) Nitrite concentrations produced at 24 h posttreatment with different concentrations of SNAP and NAP. (**C**) Cell viability, as determined by MTT assays. The mean values from two experiments are indicated. Source: [21].

**Figure 7 medsci-10-00003-f007:**
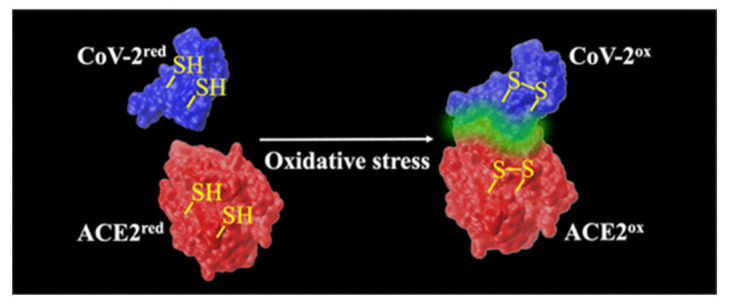
Disulfide bond forming on COVID-19 spike protein and ACE2 during oxidative stress. Both ACE2 and CoV-2 possess four disulfide bridges, equaling 8 cysteine residues. Source: [82].

**Figure 8 medsci-10-00003-f008:**
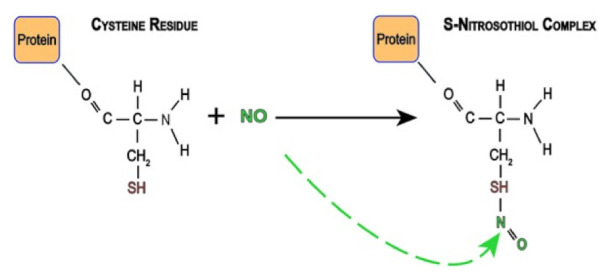
Reaction of nitric oxide with the cysteine sulfhydryl group. Source: [85].

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
