# Peer review of "Nitric Oxide: The Missing Factor in COVID-19 Severity?†"

_medsci, 2021, doi:10.3390/medsci10010003_

Round 1

Reviewer 1 Report

Please find attached a pdf file with corrections. The title must be rewritten, because "the missing link" expresses a force (of the "link") perhaps greater than what really exists. The paper is too long. I would prefer that the chapters related to the effects of NO on metabolism, pregnancy and other issues that are not directly related to the cardiovascular and respiratory implications of COVID-19 be deleted. Some figures related to basic knowledge are clearly unnecessary.
Many references are not properly presented. There are many mistakes and inconsistences.

Author Response

Dear Reviewer,

Thank you for your consideration of our work. A major advancement in this version of the article is that we were able to locate in the literature two refences that have been incorporated into paragraph 4.2 that support our initial hypothesis and findings about nitric oxide and Covid-19.

Regarding the request for changing the title, we have swapped the “link” for “factor”, which we hope reflects better the connection that we believe exists between NO and Covid-19. Regarding the suggestion that we omit the paragraphs about other conditions besides cardiovascular conditions, I would like to comment that this paper is a resubmission of our original paper that was submitted in June. During that time, we were asked to expand on those subjects by two of the previous reviewers and, we were also asked to expand on the role of NO and the immune system in this round of review. We agree that some parts may seem basic knowledge for someone well versed in nitric oxide science, but we still believe these sections add value and background to the manuscript for those inexperienced with nitric oxide. Regarding the errors in spelling and the incorrect graph of S-nitrosylation, we have amended all of those (with the exception of methemoglobinemia) and substituted the graph. We look forward to your criticism and suggestions.

Reviewer 2 Report

Alexandros Nikolaidis et al have provided a new manuscript where they added new and precious information and answered some of the reviewer points. Although they have not added data as evidence of the idea proposed, especially for the correlation they hypothesize between COVID severity and NO levels, I am in favor of publication in principle if authors could provide some changes.

  1. Although authors provided a paragraph about myeloid cells and NO, this is still not sufficient. Authors need to talk and provide more references for NO-mediated immunosuppression and discuss in detail the role of macrophages/myeloid cells during Covid infection. Moreover, authors need to cite recent work on NO and immunometabolism, also considering the peculiar bioenergetic characteristics of immune cells reported during Covid-19 infection.
  2. Authors use a wording style that is often too informal and colloquial. Please revise some phrases for a more scientific and professional sounding.
  3. Paragraph2. Nitric oxide and Covid-19 patient outcomes: What does the evidence suggest? does not provide any valuable information. Authors must decide to either delete it or provide some interesting information and evidence from retrospective studies tying NO levels with Covid-19 patient outcomes.
  4. Some sentences in paragraph 4.1 are not clear. Authors say first that NO can inhibit the viral proteases by nitrosation. Then they infer that mutations that cause alteration of a specific disulfide would have the same results as NO depletion. How do the authors explain this? (By NO promoting disulfide bonds in a situation of two or more vicinal cysteines?). Moreover authors mention that all the thiols within Covid-19 viral proteome are like a “sponge” sequestering all NO around. This is contrary to the idea of NO being good mediator for inhibiting the virus; unless the authors are trying to discuss the idea that NO can inhibit the viral replication in acute, but eventually its depletion is deleterious for the advancement of the disease. Authors need to explain all this better. Furthermore, authors need to contextualize how couple of billions of molecules of virus can lead to the substantial level of depletion of NO. Is this a mechanism purely biochemically related (by scavenging mediated by thiol groups) and not signaling-related (maybe derived from transcriptional/translational decrease of eNOS)? Authors need to cite other work on virology on similar matter. For sure authors can smooth their sentence by saying that the “sponge” effect (please change the wording), can “contribute” to depletion...
  5. Authors should compose a figure about the cysteines important in Covid-19 viral proteome and NO interacting with them together with relationships of NO with major viral disulfides.

Author Response

Dear Reviewer,

Thank you again for your consideration of our work. A major advancement in this version of the article is that we were able to locate in the literature two refences that have been incorporated into paragraph 4.2 that support our initial hypothesis and findings about nitric oxide and Covid-19. Below we write to address the specific comments and suggestions:

1)We have expanded the paragraph regarding the connection of nitric oxide and immune function and tried to find the connection between Covid-19 infection and immunometabolism as well as the exhaustion Covid-19 infers to the immune system.

2) We have tried to adjust the text in certain sections for a more formal style. 

3) We were able to locate two pieces of literature that seem to support our original hypothesis. The paragraph has thus been rewritten.

4) We have rewritten the part of paragraph 4.1 regarding disulfide bridges found on the SARS-CoV-2 and their possible role in NO depletion. We believe that we now explain more concisely the additional effect -S-S- formation (and it’s prevention by NO) can have on viral severity. As requested, we cited one more work of virology on NO’s ability to bond with viral cysteine residues on other viruses.

5) We believe we have covered Cysteine and Disulfide groups of major interest as well as their potential for interaction with NO. A figure containing all the cysteine and disulfide groups identified in Covid-19 and its various strains would be besides the authors capabilities given the time constrains.

We hope you find the revised edition of the manuscript satisfactory and we are looking forward to your criticism and suggestions.

Round 2

Reviewer 1 Report

I have asked the Editor for me to be relieved of  reviewing this manuscript. I have failed to shorten and thus improve the paper. Nevertheless, the authors' opinion must be determinant.

Author Response

Dear reviewer,

Thank you again for the criticism of our work, it has enabled us to procure a better manuscript.

Reviewer 2 Report

Authors will have to make sure the following corrections are added before the publication:

  • Authors need to cite recent work connecting macrophage immunometabolism and NO. This will have to be contextualized also considering the role of immunometabolism during Covid-19 infection. Please refer to recent publications.
  • Regarding paragraph 4.1, authors should try to at least give some rough calculations on how much of the concentration (nM) of NO and its species could be affected by certain amounts of circulating molecules of virus, which would explain systemic depletion of NO. Authors need to cite any related work on virology or microbiology on similar matter.
  • Authors should quickly explain quickly how nitrate levels correlate with dysfunctional eNOS (paragraph 4.2).
  • Regarding Fig 7 authors should write the number of Cys on both proteins.

Author Response

Dear reviewer,

Thank you for the criticism of our work. We have tried our best given the time and space constraints to address all of your comments and suggestions. We hope you find our manuscript now suitable for publication. Thank you again for your consideration of our work.

This manuscript is a resubmission of an earlier submission. The following is a list of the peer review reports and author responses from that submission.

Round 1

Reviewer 1 Report

The authors could complement whether relationship of NO with the  different mentioned entities (hypertention,  diabetes, obesity, pregnacy), has been studied in in vitro studies or in experimental models with some type of coranavirus or virus and, if so, show the effect.

The authors could include the relationship with NO levels with the obesity and adipose tissue 

It would be interesting to graphically observe an association of NO levels in the different entities and their correlation with the risk for covid-19, for example a heatmap

Review spaces and periods throughout the manuscript and check that all abbreviations are defined as COPD

Author Response

Dear reviewer,

Thank you for your consideration of our work. Regarding your suggestions about adding data regarding the correlation between NO levels, obesity e.t.c. we are currently unaware of any such in vitro testing. It would for sure be an exciting future prospect. 

Regarding the relationship of NO with obesity and adipose tissue, research in the field on humans is severely lacking and inconclusive. We have located articles on animal models but don't find the findings in them conclusive.

Regarding the development of a heatmap that associates the demographics of NO levels and Covid-19 cases, unfortunately we could find no data at that level that could support it.

We have reviewed the text and fixed some spelling and grammar errors

Reviewer 2 Report

Alexandros Nikolaidis et al highlight an interesting hypothesis for involvement of Nitric Oxide in development of Covid 19 disease. They make the connection based on the importance of NO levels and NOS status in the major pre-existing conditions believed to be severe predispositions for Covid hospitalizations and deaths: age, cardiovascular diseases, pregnancy, diabetes and COPD. They also provide notions on possible therapies for covid patients and nutritional remedies for improvement of systemic NO levels. While this manuscript may have a substantial impact in Covid research, it lacks major evidence that could at least partially support authors’ idea.

  1. In the abstract please modify the wording about resistance of Covid variants to available vaccines: it has not been shown that any variant is anywhere close to be resistant to developed vaccines.
  2. Please add references to the statement referring to patients still experiencing effects after acute phase.
  3. It would be important that authors add any evidence of lower NO levels in severe Covid patients vs mild/asymptomatic.
  4. When author talk about COVID-19 being able to accelerate endothelial dysfunction and NO deficiency, although the point is interesting, it represents NO levels being a consequence of Covid and not a cause, unless this is also a point authors are trying to infer. This connects with point n3: would it be hard to assess NO levels in Covid cases as lower NO could be a consequence of lung disease for example? In this case are there retrospective studies authors could utilize to validate their point of “predisposition” to severe Covid when NO levels are lower or NOS enzymes are altered?
  5. Authors could merge the paragraph about heart disease with hypertension.
  6. About nitric oxide and asthma, it would be important that authors explain the counter intuitive data they cite about enhancing plasma NO production/availability via eNOS upregulation can reduce asthma severity while exhaled NO is already a characteristic of asthma. Please add support mechanisms for these statements.
  7. Considering the importance of NOS in immune system and of NOS2 in immunometabolism, it would be mandatory that authors include a paragraph about myeloid cells and macrophages specifically, taking into account how fundamental is the role of these cells for Covid disease stages and resolution.
  8. It is not clear the sentence about the role of nitroglycerine and other drugs on thiol groups availability. Please provide organic explanation for this.
  9. It would be important to provide references for the doses cited for intake of nitrate and folate for nutritional supplementation. Moreover authors should explain how folate in combination with nitrate is important for high risk Covid people.

Author Response

Dear reviewer,

Thank you for your consideration of our work and your valuable insights and criticism. I will try to reply to your suggestions one by one

  • In the abstract please modify the wording about resistance of Covid variants to available vaccines: it has not been shown that any variant is anywhere close to be resistant to developed vaccines.
  • Agreed. Abstract was modified accordingly
  • Please add references to the statement referring to patients still experiencing effects after acute phase.
  • Added reference
  • It would be important that authors add any evidence of lower NO levels in severe Covid patients vs mild/asymptomatic.
  • Unfortunately we could not locate any data regarding NO levels of asymptomatic Covid-19 patients
  • When author talk about COVID-19 being able to accelerate endothelial dysfunction and NO deficiency, although the point is interesting, it represents NO levels being a consequence of Covid and not a cause, unless this is also a point authors are trying to infer.
  • In our opinion low NO levels are both a risk factor for Covid-19 serious illness and a consequence of Covid-19 illness
  • This connects with point n3: would it be hard to assess NO levels in Covid cases as lower NO could be a consequence of lung disease for example?
  • The simplest way to access NO levels in the plasma is by measuring Nitrate/Nitrite(end products of NO metabolism) levels in the plasma. Strips that measure nitrate/nitrite levels in the saliva exist but their accuracy has been a matter of scientific debate. Regarding NO levels in asthmatic patients, machines that measure them in exhaled air exist. 
  • In this case are there retrospective studies authors could utilize to validate their point of “predisposition” to severe Covid when NO levels are lower or NOS enzymes are altered?
  • Not yet. We would be glad to participate in such studies.
  • Authors could merge the paragraph about heart disease with hypertension.
  • Agreed and done
  • About nitric oxide and asthma, it would be important that authors explain the counter intuitive data they cite about enhancing plasma NO production/availability via eNOS upregulation can reduce asthma severity while exhaled NO is already a characteristic of asthma. Please add support mechanisms for these statements.
  • We are afraid that would get us over the limit for number of pages and references. One reviewer has asked us to "drastically cut" our manuscript.
  • Considering the importance of NOS in immune system and of NOS2 in immunometabolism, it would be mandatory that authors include a paragraph about myeloid cells and macrophages specifically, taking into account how fundamental is the role of these cells for Covid disease stages and resolution.
  • Same consideration for number of pages and references applies. We will try to address these in future work.
  • It is not clear the sentence about the role of nitroglycerine and other drugs on thiol groups availability. Please provide organic explanation for this.
  • Modified the text to make it clearer
  • It would be important to provide references for the doses cited for intake of nitrate and folate for nutritional supplementation. Moreover authors should explain how folate in combination with nitrate is important for high risk Covid people.
  • Added with references for support

Reviewer 3 Report

The authors do not attempt to address the double side of NO. In other words, it seems to me that they only show the aspects, the clues, that can support their theory, which in fact is quite interesting, even suggestive for me, forgetting, perhaps, other alternative scenarios. Also, the manuscript should be drastically shortened.

Author Response

Dear reviewer,

Thank you for your consideration of our work. Regarding your comment about the "double side of NO", I assume you refer to the negative health implications of NO generated from iNOS and nNOS. We are unable to address this without adding many pages and references. We have included a paragraph explaining the manuscript's limitations. Regarding your comment that we "drastically cut" our manuscript, two of the other reviewers have asked for more data and references supporting it. We decided to comply, though we are still waiting for input from the editor. As for the existence of alternative scenarios, we will be more than happy to explore them with the rest of the scientific community once our manuscript gets published. 

Round 2

Reviewer 2 Report

Alexandros Nikolaidis et al have provided a new manuscript that barely differs from the original submission. They have not addressed many of the reviewer’s comments and still haven’t added data as evidence of the idea proposed. I am not in favor of publication unless authors can revise according to all the un-answered points from the first review.

Author Response

Dear reviewer,

Thank you for your criticism of our work. The following changes have been made to the new manuscript we will upload in response to your criticism:

  • In the abstract please modify the wording about resistance of Covid variants to available vaccines: it has not been shown that any variant is anywhere close to be resistant to developed vaccines.

Agreed. The abstract was modified accordingly.

  • Please add references to the statement referring to patients still experiencing effects after acute phase.

Reference to the CDC’s website regarding post-covid 19 syndrome was added

  • It would be important that authors add any evidence of lower NO levels in severe Covid patients vs mild/asymptomatic.

Unfortunately we could not locate any data regarding NO levels of asymptomatic Covid-19 patients. It would be an exciting future field of research.

  • When author talk about COVID-19 being able to accelerate endothelial dysfunction and NO deficiency, although the point is interesting, it represents NO levels being a consequence of Covid and not a cause, unless this is also a point authors are trying to infer.

In our opinion low NO levels are both a risk factor for Covid-19 serious illness and a consequence of Covid-19 illness. Among other things, low NO levels could negatively impact antibodies ability to stop the virus’s spike protein from latching onto the human cells. We have tried to make this double relationship clearer in the newest version of our manuscript.

  • This connects with point n3: would it be hard to assess NO levels in Covid cases as lower NO could be a consequence of lung disease for example?

The simplest way to access NO levels in the plasma is by measuring Nitrate/Nitrite(end products of NO metabolism) levels in the plasma. Strips that measure nitrate/nitrite levels in the saliva exist but their accuracy has been a matter of scientific debate. Regarding NO levels in asthmatic patients, machines that measure them in exhaled air exist. It has been observed that asthmatics, especially children, exhale higher amounts of NO. Such amounts however are in the ppb concentration, highly lower than the ppm amounts currently used in therapeutics.

  • In this case are there retrospective studies authors could utilize to validate their point of “predisposition” to severe Covid when NO levels are lower or NOS enzymes are altered?

Not yet. We would be glad to participate in such studies.

  • Authors could merge the paragraph about heart disease with hypertension.

We agree with this suggestion and changed the manuscript thus.

  • About nitric oxide and asthma, it would be important that authors explain the counter intuitive data they cite about enhancing plasma NO production/availability via eNOS upregulation can reduce asthma severity while exhaled NO is already a characteristic of asthma. Please add support mechanisms for these statements.

We are afraid that would get us over the limit for number of pages and references. One reviewer has asked us to "drastically cut" our manuscript. As explained in the text, higher exhaled NO levels in asthmatics are a result of iNOS activity, not eNOS.

  • Considering the importance of NOS in immune system and of NOS2 in immunometabolism, it would be mandatory that authors include a paragraph about myeloid cells and macrophages specifically, taking into account how fundamental is the role of these cells for Covid disease stages and resolution.

A paragraph discussing the role of Nitric Oxide in immune suppressed populations has been added which also addresses the reviewers inquiries about its role on myeloid cells and macrophages

  • It is not clear the sentence about the role of nitroglycerine and other drugs on thiol groups availability. Please provide organic explanation for this.

We have modified the text accordingly hoping to make the role of nitroglycerin on thiol depletion clearer. Also added a reference to a MD article that elucidates the successful utilization of sulfhydryl donors to combat the phenomenon.

  • It would be important to provide references for the doses cited for intake of nitrate and folate for nutritional supplementation. Moreover authors should explain how folate in combination with nitrate is important for high risk Covid people.

Added references explaining the role of dietary nitrate as a NO precursor/storage in the body as well as folic acid’s ability to restore eNOS function

Reviewer 3 Report

Provided that the suggestions I sent last week (in an attached pdf document) have not been incoprporated to the manuscript, I can't give my acceptation.

Author Response

Dear reviewer,

thank you for our criticism of our work. Regarding your request to address the "other side of NO" we have added references regarding the implication of NO in the development of gastrointestinal cancer. Regarding your request to "drastically cut" our subject, we will have to politely decline. The other 2 reviewers have asked us for further elucidation and references and we chose to comply.

Round 3

Reviewer 3 Report

I find that the authors' response is far from agreeing  with the suggestions issued by myself and other referee/-s.